# Lasianosides F–I: A New Iridoid and Three New Bis-Iridoid Glycosides from the Leaves of *Lasianthus verticillatus* (Lour.) Merr.

**DOI:** 10.3390/molecules25122798

**Published:** 2020-06-17

**Authors:** Gadah Abdulaziz Al-Hamoud, Raha Saud Orfali, Yoshio Takeda, Sachiko Sugimoto, Yoshi Yamano, Nawal M. Al Musayeib, Omer Ibrahim Fantoukh, Musarat Amina, Hideaki Otsuka, Katsuyoshi Matsunami

**Affiliations:** 1Department of Pharmacognosy, College of Pharmacy, King Saud University, Riyadh 11495, Saudi Arabia; galhamoud@ksu.edu.sa (G.A.A.-H.); rorfali@ksu.edu.sa (R.S.O.); nalmusayeib@ksu.edu.sa (N.M.A.M.); ofantoukh@ksu.edu.sa (O.I.F.); mamina@ksu.edu.sa (M.A.); 2Graduate School of Biomedical and Health Sciences, Hiroshima University, 1-2-3 Kasumi, Minami-ku, Hiroshima 734-8553, Japan; ssugimot@hiroshima-u.ac.jp (S.S.); yamano@hiroshima-u.ac.jp (Y.Y.); 3Faculty of Integrated Arts and Sciences, The University of Tokushima, 1-1 Minamijosanjima-Cho, Tokushima 770-8502, Japan; takeda@ias.tokushima-u.ac.jp; 4Faculty of Pharmacy, Yasuda Women’s University, 6-13-1 Yasuhigashi, Asaminami-ku, Hiroshima 731-0153, Japan; otsuka-h@yasuda-u.ac.jp

**Keywords:** Rubiaceae, iridoids, bis-iridoids, *Lasianthus verticillatus*, lasianoside

## Abstract

A series of iridoid glycosides were isolated from the leaves of *Lasianthus verticillatus* (Lour.) Merr., belonging to family Rubiaceae. A new iridoid glycoside, lasianoside F (**1**), and three new bis-iridoid glycosides, lasianosides G–I (**2**–**4**), together with four known compounds (**5**–**8**) were isolated. The structures were established by spectroscopic methods, including 1D and 2D NMR experiments (^1^H, ^13^C, DEPT, COSY, HSQC, HMBC, and NOESY) in combination with HR-ESI-MS and CD spectra.

## 1. Introduction

Rubiaceae is the fourth-largest angiosperm family, comprising approximately 660 genera and 11,500 species and classified into 42 tribes [1]. Rubiaceae has a long history of investigation on the distribution of iridoid glycoside through its species. These investigations were started by isolation of asperuloside from six plants belonging to the family Rubiaceae, as a characteristic iridoid for this genus [2]. The classification of the occurrence of iridoid glucoside in Rubiaceae subfamilies was initiated by Kooiman 1969 [3]. Later, this classification was approved by investigation of 35 selected Rubiaceae plants by TLC, GC, and GC-MS; the result revealed that asperuloside and deacetylasperulosidic acid occur in most plants of the Rubioideae subfamily, especially in *Lasianthus* species [4]. Previous phytochemical studies on some *Lasianthus* species revealed the presence of iridoids, iridoid glycosides, anthraquinones, and terpenes [5,6,7,8,9,10]. In our previous study, we isolated a bis-iridoid glycoside from *L. wallichii* for the first time [11] and five undescribed iridane type glycosides, lasianosides A–E, from *L. verticillatus* [12]. These results indicated that the genus *Lasianthus* is a promising rich source in secondary metabolites; however, only limited numbers of *Lasianthus* species have been investigated until now. To continue research of this genus, we performed further phytochemical investigation of the leaves of *L. verticillatus*. As a result, a new iridoid glycoside, lasianoside F (**1**), and three new bis-iridoid glycosides, lasianosides G–I (**2**–**4**) were isolated in this study. The chemical structures were determined by spectroscopic (Appendix A) and chemical analyses, as shown in Figure 1.

## 2. Results

### 2.1. Isolation and Spectroscopic Analyses of the Compounds

The 1-BuOH and EtOAc fractions of methanolic extract of the leaves of *L. verticillatus* were subjected to fractionation by Diaion HP-20 and silica gel column chromatographies, respectively. The resulting fractions were separated on octadecylsilane (ODS) column chromatography, then purified by preparative high-performance liquid chromatography (HPLC) to obtain a new iridoid glycoside (**1**), three new bis-iridoid glycosides (**2**–**4**), in addition to five known compounds: asperuloside (**5**), deacetyl asperuloside (**6**) [13], besperuloside (**7**) [14], and iridoid glycoside dimer (**8**) [6] (Figure 1).

#### 2.1.1. Chemical Structure of Compound **1**

Compound (**1**) was obtained as a colorless amorphous powder with a specific optical rotation of [α] ^22^_D_ − 65.5. The molecular formula was deduced to be C_21_H_28_O_11_ from HR-ESI-MS (*m*/*z* 479.1521 [M + Na]^+^, calcd for C_21_H_28_O_11_Na, 479.1524), which suggested eight degrees of unsaturation. The UV spectrum showed absorption maxima at 234 nm, indicating the presence of an enone system, and IR absorption bands at 3406, 1733, 1658, and 1634 cm^−1^ that corresponded to hydroxy, carbonyl, and olefinic groups. The ^1^H-NMR spectrum of **1** (Table 1) showed one oxymethylene at *δ*_H_ 4.69 and 4.81 ppm, two olefinic protons; one at *δ*_H_ 7.32 ppm assigned to conjugated enol ether and the other at *δ*_H_ 5.75 ppm, two methines at *δ*_H_ 3.70 and 3.31 ppm, two oxymethines at *δ*_H_ 5.59 and *δ*_H_ 5.98 ppm, one anomeric proton at *δ*_H_ 4.70, together with signals of isovaleroyl unit (one methylene at *δ*_H_ 2.27ppm, one methine at *δ*_H_ 2.09 ppm, and two equivalent methyl signals at *δ*_H_ 0.98 ppm). The ^13^C-NMR spectrum of **1** showed 21 signals, of which six signals could be attributed to a glucopyranosyl unit (*δ*c 100.0, 78.4, 77.9, 74.7, 71.6, and 62.8), ten signals for an iridoid skeleton (*δ*c 37.5, 45.4, 61.7, 86.4, 93.3, 106.2, 129.1, 144.4, 150.4, and 172.6) which were similar to those reported for asperuloside (**5**) [13], and five other signals that contributed to the isovaleroyl unit (*δ*c 22.8 (2C), 26.8, 44.0, 174.2). 

The HMBC correlations (Figure 2) from H_2_-10 (*δ*_H_ 4.69 and 4.81) to C-1ʺ (*δ*c 174.1), and from anomeric proton H-1ʹ (*δ*_H_ 4.70) to C-1 (*δ*c 93.3) ascertained the presence of isovaleroyl moiety on C-10 and glucosyl moiety on C-1, respectively. The coupling constant of anomeric proton H-1ʹ (*J* = 7.9 Hz) indicated β linkage for glucose moiety, while acid hydrolysis of **1** yielded D-glucose that was identified by HPLC analysis with a chiral detector in comparison with authentic D-glucose. The relative configuration of **1** was assigned on the basis of a NOESY experiment (Figure 3). The correlations observed between H-5/H-6 and H-9 suggested β-orientation of H-5, H-6, and H-9. The presence of the correlations of H-1/H-9 and H-1/H-10, and the absence of H-1/H-5 were in good agreement with the proposed structure. The chemical shift values and the coupling patterns of **1** were similar to those of asperuloside (**5**) [13]. The CD spectrum (∆ε= −4.11 at 245 nm) confirmed the absolute configuration of **1** to be the same as asperuloside (**5**). Thus, compound **1** was identified as isovalerate of deacetyl asperuloside, designated as lasianoside F.

#### 2.1.2. Chemical Structure of Compound **2**

Compound (**2**) was isolated as a colorless amorphous powder with a specific optical rotation of [α] ^22^_D_ − 55.0. Its molecular formula C_36_H_44_O_22_, from its HR-ESI-MS (*m*/*z* 851.2214 [M + Na]^+^ (calcd for C_36_H_44_O_22_Na 851.2216), indicating 15 degrees of unsaturation. The UV spectra of **2** exhibited absorption maxima at 236 nm, characteristic of an enol ether system. Similarly, IR spectra displayed absorption bands corresponding to hydroxy, carbonyl, and olefinic groups at 3309, 1736, 1541 cm^−1^, respectively. Duplication of the signals in both ^1^H and ^13^C-NMR spectra (Table 2 and Table 3) of **2** clearly implied the dimeric nature of two iridoid glycosides. The proton signals arising in the region of δ_H_ 3.27–4.92 in ^1^H-NMR spectrum including two anomeric protons at δ_H_ 4.70 (1H, d, *J* = 7.8 Hz) and 4.92 (1H, d, *J* = 8.2 Hz) supported the presence of two glucosyl units in **2** (Table 2). Furthermore, two sp^2^ methine proton signals at δ_H_ 7.15 (1H, d, *J* = 1.9 Hz) and 7.70 (1H, d, *J* = 1.1 Hz), which are characteristic for C-3 protons confirmed the presence of two iridoid moieties having an enol ether function. Consistent with these observations, the ^13^C-NMR spectrum showed 36 signals comprising four carbonyl carbons (δc 167.6, 172.2, 172.2 and 172.7), eight olefinic carbons (δc 106.3, 107.7, 129.1, 131.6, 143.9, 146.5, 150.1 and 156.3), six oxygenated carbons (four methines δc 75.2, 86.1, 94.1 and 101.8; two methylenes δc 61.9 and 63.8), four sp^3^ methine carbons (δc 37.6, 42.9, 45.1 and 45.9) together with two anomeric carbons (δc 98.7, 100.9) and oxygenated carbons arising in the region of δc 62.7–78.6 belonging to two glucose moieties (Table 3). Thus, the two partial structures of **2** were referred to as units “A” and “B” and determined to be asperuloside (**5**) and asperulosidic acid [5,13], respectively. The 1D and 2D-NMR data of compound **2** were very similar to those of the bis-iridoid glucoside (**8**) that reported in [11]. The only evident difference was observed in the glycosyl part of unit “A”, i.e., the lower field shifted H-2ʹ at δ_H_ 4.80, and upper field shifted H_2_-6ʹ, δ_H_ 3.69 and 3.94 ppm, indicating that **2** was a positional isomer of **8**, and the attachment site between “A” and “B” units was deduced to be at C-2ʹ of unit “A” via an ester linkage. This assumption was further verified by a correlation from H-2ʹ (δ_H_ 4.80) of unit “A” to C-11 (δc 167.6) of unit “B” in the HMBC spectrum (Figure 2). Moreover, acid hydrolysis of **2** gave D-glucose, which was identified by HPLC analysis with a chiral detector, while β-anomeric configurations were established from the coupling constant of anomeric protons, 8.2 and 7.8 Hz. The relative and absolute configurations of aglycone parts of **2** were determined to be identical to **5** by comparison of their chemical shift values, coupling constants, NOESY experiment (Figure 3), and CD data. Therefore, the structure of **2** was characterized as shown in Figure 1, named lasianoside G.

#### 2.1.3. Chemical Structure of Compound **3**

Compound (**3**) was obtained as a colorless amorphous powder, with a specific optical rotation of [α] ^22^_D_ − 59.9. The molecular formula was assigned as C_36_H_44_O_22_ by HR-ESI-MS at *m*/*z* 851.2212 [M + Na]^+^ (calcd for C_36_H_44_O_22_ Na 851.2216), indicating that **3** was also another positional isomer of **8**. Comparison of ^1^H and ^13^C-NMR data (Table 2 and Table 3) showed that the structure of **3** was similar to that of **8**. The significant change occurred in glucose moiety of unit “A”, i.e., the chemical shift of H-3ʹ moved to downfield at *δ*_H_ 5.08 ppm, and the chemical shift of H_2_-6ʹ moved to upfield at *δ*_H_ 3.74 and 3.95 ppm, which suggested that the position of esterification between unit “A” and “B” was changed from H-6ʹ to H-3ʹ. This suggestion was supported by a correlation between the H-3ʹ (*δ*_H_ 5.08) of unit “A” and C-11(*δ*c 168.6) of unit “B” in HMBC spectrum (Figure 2). The structure of this compound was verified by further analysis of 2D-NMR data, including COSY, HSQC, and HMBC spectra. The relative and absolute configurations of aglycone parts of **3** were identical to those of **2** by comparison of their chemical shift values, coupling constants, NOESY experiment (Figure 3), and CD analysis. From these data, the structure of **3** was characterized as shown in Figure 1, designated as lasianoside H. 

#### 2.1.4. Chemical Structure of Compound **4**

Compound (**4**) was isolated as a colorless amorphous powder, with a specific optical rotation of [α]^22^_D_ − 60.1. It has molecular formula of C_36_H_44_O_22_ established from its ^13^C-NMR data and positive mode HR-ESI-MS [*m*/*z* 851.2215 [M + Na]^+^ (calcd for C_36_H_44_O_22_ Na 851.2216)]. The ^13^C-NMR data showed signals resembling those of **8**, except the presence of two *sp*^3^ methines, C-4 at *δ*c 44.4 and C-3 at 97.4 ppm in unit “B” of **4**, instead of resonances of two olefinic carbons at the same position of **8**, in addition to lower field shift of C-11 and C-6 to *δ*c 176.9 and 87.9 ppm, respectively (Table 4). This change coincided with the disappearance of an enol ether proton signal and the appearance of methine proton at *δ*_H_ 3.36 ppm together with oxymethine proton at *δ*_H_ 5.27 ppm that correspond to H-4 and H-3 of unit “B”, respectively (Table 4). The above data suggested the absence of a double bond between C-3 and C-4 and the presence of *γ*-lactone ring in the aglycone part of unit “B”. The occurrence of γ-lactone was confirmed by HMBC correlation from H-6 (*δ*_H_ 5.41) to C-11 (*δ*c 176.9) (Figure 2). A detailed analysis of NMR data (COSY, HSQC, and HMBC) suggested two partial structures in **4**, i.e., asperuloside (**5**) [13] and 3,4-dihydro-3-oxy asperuloside [15]. The attachment between “A” and “B” units was found to be between C-6ʹ of unit “A” and C-3 of unit “B” via *O*-linkage due to a long-range correlation between H_2_-6ʹ of unit “A” (*δ*_H_ 3.95 and 4.18) and C-3 of unit “B” (*δ*c 97.4) in the HMBC spectrum (Figure 2). HPLC analysis after acid hydrolysis of **4** revealed that the glycosyl units were D-configurations. The relative and absolute configurations of unit “A” were the same as **5** by comparison of NOESY, chemical shifts, and coupling constants. On the other hand, the stereochemistry of part “B” was achieved by NOESY analysis, particularly for those of chiral centers H-4, H-5, H-6, and H-9. In the NOESY spectrum, the correlations between H-5/H-4, H-6, and H-9, indicated β-orientation of H-4, H-5, H-6, and H-9 (Figure 3). The stereochemistry of C-3 in unit “B” was also determined as Figure 1, because of the chemical shift similarity with 3,4-dihydro-3-methoxy asperuloside [15], coupling constants, and the absence of NOE correlation between H-3/H-4,5,9. The CD spectrum showed essentially the same cotton effect as asperuloside (**5**). Base on the above findings, the structure of **4** was assigned as shown in Figure 1, named lasianoside I.

## 3. Materials and Methods 

### 3.1. General Methods

Optical rotations and CD data were measured with JASCO P-1030 and Jasco J-720 polarimeters (Jasco, Tokyo, Japan), respectively. IR spectra were recorded on Horiba FT-710 Fourier transform infrared (Horiba, Kyoto, Japan), and UV spectra were obtained on Jasco V-520 UV/Vis spectrophotometers. NMR measurements were performed on Bruker Avance 500 and 700 spectrometers, with tetramethylsilane (TMS) as internal standard (Bruker Biospin, Rheinstetten, Germany). Stable conformations were calculated using a Merck Molecular Force Field (MMFF94s). HR-ESI-MS spectra were obtained using LTQ Orbitrap XL mass spectrometer (Thermo Fisher Scientific, Waltham, MA). Diaion HP-20 (Atlantic Research Chemical Ltd., UK), silica gel 60 (230–400 mesh, Merck, Germany), and octadecyl silica (ODS) gel (Cosmosil 75C_18_–OPN (Nacalai Tesque, Kyoto, Japan; Φ = 35 mm, L = 350 mm) were used for column chromatography (CC). Analytical thin-layer chromatography (TLC) was performed on precoated silica gel plates 60 GF_254_ (0.25 mm in thickness, Merck). For visualization of TLC plates, 10% sulfuric acid reagent was used. Isolated compounds were purified by HPLC using an ODS column (Cosmosil 10C_18_-AR, Nacalai Tesque, Kyoto, 10 mm × 250 mm, flow rate 2.5 mL/min) with a mixture of H_2_O and MeOH and the eluate was monitored by refractive index and/or a UV detector. After hydrolysis, the sugars were analyzed by HPLC using an amino column (Shodex Asahipak NH2P-50 4E (4.6 mm × 250 mm), CH_3_CN-H_2_O (3:1) 1mL/min) together with a chiral detector (Jasco OR-2090plus).

### 3.2. Plant Material

Leaves of *L. verticillatus* were collected in 2000 from Iriomote Island, Okinawa Prefecture, Japan. A voucher specimen of the plant was deposited in the herbarium of the Department of Pharmacognosy, Faculty of Pharmaceutical Sciences, Hiroshima University (IR0009-LT).

### 3.3. Extraction and Isolation

The air-dried and powdered leaves (7.0 kg) of *L. verticillatus* (Lour.) Merr. were extracted by maceration with MeOH (98 L × 2) and concentrated to 90% MeOH solution, then defatted with 3 L of *n*-hexane. The remaining solution was evaporated and resuspended in 1 L H_2_O and extracted by EtOAc (1 L × 3, 46.5 g) and 1-BuOH (1 L × 3, 178.5 g), successively.

A portion of 1-BuOH fraction (124.5 g) was fractionated by Diaion HP-20 column (Φ = 10 cm, L = 60 cm, 2.5 kg), eluting with stepwise MeOH/H_2_O gradient (0 to 60% MeOH, 15 L each); similar fractions were grouped together to give 20 fractions (Fr. Lt1–Lt20). The fraction Lt8 (18.8 g) was separated on silica gel CC (Φ = 4.5 cm, L = 50 cm, 400 g), eluting with CHCl_3_ / MeOH gradient (100:0 to 70:30, 2.5 L each) to obtain 16 fractions (Fr. Lt8.1–Lt8.16). Fractions Lt8.13 (240 mg) was subjected to open reversed phase (ODS) CC with 10% aq. methanol (400 mL) to 100% methanol (400 mL), linear gradient, lead to six fractions (Frs. Lt8.13.1–Lt8.13.6). Purification of Lt8.13.2 (174 mg) by preparative HPLC, 5% aq. methanol, to give compound **6** (30.5 mg). Fraction Lt15 (7.22 g) was proceeded on silica gel CC (Φ = 5.2 cm, L = 38 cm, 350 g), using the gradient mixture of CHCl_3_/MeOH (100:0 to 70:30, 2.5 L each), to obtain 12 fractions (Frs. Lt15.1–Lt15.12). The residue Lt15.8 (1.88 g) was separated by HPLC, 40% aq. methanol, to provide compounds **8** (42.0 mg) and **2** (31.0 mg). The fraction Lt17 (6.15 g) was further purified by silica gel CC (Φ = 5 cm, L = 40 cm, 380 g), eluting with stepwise CHCl_3_/MeOH gradient (100:0 to 70:30, 2.4 L each), to obtain 11 fractions (Frs. Lt17.1–Lt17.11). The residue Lt17.5 (282 mg) was further purified by HPLC, 25% aq. acetone, to give compound **1** (13.4 mg), while the other residue Lt17.7 (839 mg) was separated by HPLC, 28% aq. acetone, to obtain compounds **3** (25.0mg) and **4** (13.0). A portion of EtOAc fraction (42.8 g) was chromatographed on silica gel CC (Φ = 5 cm, L = 40 cm, 400 g), eluting with CHCl_3_ (2.5 L), followed by stepwise CHCl_3_/MeOH (50:1, 20:1, 15:1, 10:1, 7:1, 5:1, 3:1, 1:1, 2.5 L each), then 100% MeOH (2.5 L), lead ten fractions (Frs. LtE1–LtE10). Each fraction of LtE4 (21.9 g) and LtE6 (22.0 g) was separated by open reversed phase (ODS) CC with 10% aq. methanol (400 mL) to 100% methanol (400 mL), linear gradient, lead eight fractions (Frs. LtE4.1–LtE4.8 and Frs. LtE6.1–LtE6.8, respectively). The residue LtE4.2 (199 mg) was further purified by HPLC, 20% aq. acetone, to provide compound **5** (7.60 mg), while the residue LtE6.4 (262 mg) was purified by HPLC, 35% aq. acetone, to give **7** (5.60 mg).

### 3.4. Spectroscopic Data of Compounds ***1***–***4***

Lasianoside F (**1**): (2a*S*,4a*S*,5*S*,7b*S*)-4-[(3-methylbutanoyloxy)methyl]-5-(β-D-glucopyranosyloxy)-2a,4a,5,7b-tetrahydro-1H-2,6-dioxacyclopent[cd]inden-1-one. Colorless amorphous powder [α]^22^_D_ − 65.5 (c 0.88, MeOH); HR-ESI-MS (positive ion mode): *m*/*z*: 479.1521 [M + Na]^+^ (calcd for C_21_H_28_O_11_Na, 479.1524); CD λ_max_ (c 2.19 × 10^−5^ M, MeOH) nm (∆ε): 245 (−4.11); UV (MeOH) λ_max_ nm (log ε) 234 (4.04); IR (film) ν_max_: 3406, 2960, 1733, 1658, 1634, 1292, 1183, 1164, 1077, 1017, 762 cm^−1^; ^1^H-NMR (500 MHz, CD_3_OD) and ^13^C (175 MHz, CD_3_OD): Table 1.

Lasianoside G (**2**): (2a*S*,4a*S*,5*S*,7b*S*)-4-[(acetyloxy)methyl]-5-[[2-*O*-[[(1*S*,4a*S*,5*S*,7a*S*)-7-[(acetyloxy)methyl]-1-(β-D-glucopyranosyloxy)-1,4a,5,7a-tetrahydro-5-hydroxycyclopenta[c]pyran-4-yl]carbonyl]-β-D-glucopyranosyl]oxy]-2a,4a,5,7b-tetrahydro-1H-2,6-dioxacyclopent[cd]inden-1-one. Colorless amorphous powder [*α*]^22^_D_ − 55.0 (c 0.10, MeOH); HR-ESI-MS (positive ion mode): *m*/*z*: 851.2214 [M + Na]^+^ (calcd for C_36_H_44_O_22_Na, 851.2216); CD λ_max_ (c 2.35 × 10^−5^ M, MeOH) nm (∆ε): 235 (−8.04); UV (MeOH) λ_max_ nm (log ε) 236 (4.10); IR (film) ν_max_: 3309, 2925, 1736, 1716, 1541, 1260, 1162, 1057, 1033, 669 cm^−1^; ^1^H-NMR (500 MHz, CD_3_OD) and ^13^C (175 MHz, CD_3_OD): Table 2 and Table 3.

Lasianoside H (**3**): (2a*S*,4a*S*,5*S*,7b*S*)-4-[(acetyloxy)methyl]-5-[[3-*O*-[[(1*S*,4a*S*,5*S*,7a*S*)-7-[(acetyloxy)methyl]-1-(β-D-glucopyranosyloxy)-1,4a,5,7a-tetrahydro-5-hydroxycyclopenta[c]pyran-4-yl]carbonyl]-β-D-glucopyranosyl]oxy]-2a,4a,5,7b-tetrahydro-1H-2,6-dioxacyclopent[cd]inden-1-one. Colorless amorphous powder [*α*]^22^_D_ − 59.9 (c 1.38, MeOH); HR-ESI-MS (positive ion mode): *m*/*z*: 851.2212 [M + Na]^+^ (calcd for C_36_H_44_O_22_Na, 851.2216); CD λ_max_ (c 1.41 × 10^−5^ M, MeOH) nm (∆ε): 245 (−5.73); UV (MeOH) λ_max_ nm (log ε) 235 (4.31); IR (film) ν_max_: 3388, 2932, 1730, 1658, 1632, 1261, 1158, 1075, 1044, 788 cm^−1^; ^1^H-NMR (500 MHz, CD_3_OD) and ^13^C (175 MHz, CD_3_OD): Table 2 and Table 3.

Lasianoside I (**4**): (2a*S*,4a*S*,5*S*,7b*S*)-4-[(acetyloxy)methyl]-5-[[6-*O*-[[(2a*R*,4a*S*,5*R*,7*S*,7a*S*,7b*S*)-4-[(acetyloxy)methyl]-5-(β-D-glucopyranosyloxy)-2a,4a,5,7,7a,7b-hexahydro-1H-2,6-dioxacyclopenta[cd]inden-1-one-7-yl]]-β-D-glucopyranosyl]oxy]-2a,4a,5,7b-tetrahydro-1H-2,6-dioxacyclopent[cd]inden-1-one. Colorless amorphous powder [α]^22^_D_ − 60.1 (c 1.38, MeOH); HR-ESI-MS (positive ion mode): *m*/*z*: 851.2215 [M + Na]^+^ (calcd for C_36_H_44_O_22_Na, 851.2216); CD λ_max_ (c 1.14 × 10^−5^ M, MeOH) nm (∆ε): 245 (−3.48); UV (MeOH) λ_max_ nm (log ε) 234 (4.23); IR (film) ν_max_: 3407, 2927, 1739, 1658, 1254, 1175, 1070, 1052, 1017, 758 cm^−1^; ^1^H-NMR (500 MHz, CD_3_OD) and ^13^C (175 MHz, CD_3_OD): Table 4.

Asperuloside (**5**): Colorless amorphous powder, [α]^24^_D_ − 170.6 (c 0.32, MeOH); HR-ESI-MS (positive ion mode): *m*/*z*: 437.1053 [M + Na]^+^ (calcd for C_18_H_22_O_11_Na, 437.1054); ^13^C-NMR (175 MHz, CD_3_OD) *δ*_C_: 20.6 (*C*H_3_-CO-), 37.4 (C-5), 45.3 (C-9), 61.9 (C-10), 62.8 (C-6ʹ), 71.6 (C-4ʹ), 74.6 (C-2ʹ), 77.9 (C-5ʹ), 78.4 (C-3ʹ), 86.3 (C-6), 93.3 (C-1), 100.0 (C-1ʹ), 106.2 (C-4), 128.9 (C-7), 144.3 (C-8), 150.3 (C-3), 172.3 (C-11), 172.6 (CH_3_-CO-).

Deacetyl asperuloside (**6**): Colorless amorphous powder, [α]^24^_D_ − 125.4 (c 0.62, MeOH); HR-ESI-MS: *m*/*z*: 395.0946 [M + Na]^+^ (calcd for C_16_H_20_O_10_Na 395.0948); ^13^C-NMR (175 MHz, CD_3_OD) *δ*_C_: 37.5 (C-5), 45.0 (C-9), 60.1 (C-10), 62.8 (C2-6ʹ), 71.6 (C-4ʹ), 74.6 (C-2ʹ), 77.9 (C-5ʹ), 78.4 (C-3ʹ), 86.7 (C-6), 93.3 (C-1), 99.3 (C-1ʹ), 106.5 (C-4), 125.7 (C-7), 149.8 (C-8), 150.3 (C-3), 172.9 (C-11).

Besperuloside (**7**): Colorless amorphous powder, [α]^25^_D_ − 109.8 (c 0.38, MeOH); HR-ESI-MS: *m*/*z*: 499.1210 [M + Na]^+^ (calcd for C_23_H_24_O_11_Na 499.1211); ^13^C-NMR (175 MHz, CD_3_OD) *δ*_C_: 37.4 (C-5), 45.0 (C-9), 62.6 (C-10), 62.7 (C-6ʹ), 71.5 (C-4ʹ), 74.6 (C-2ʹ), 77.9 (C-5ʹ), 78.4 (C-3ʹ), 86.3 (C-6), 93.4 (C-1), 100.0 (C-1ʹ), 106.2 (C-4), 129.4 (C-7), 129.7 (C-3ʹʹ, 5ʹʹ), 130.7 (C-2ʹʹ, 6ʹʹ), 130.9 (C-1ʹʹ), 134.6 (C-4ʹʹ), 144.3 (C-8), 150.3 (C-3), 165.9 (C-7ʹʹ), 172.9 (C-11).

Compound (**8**): Colorless amorphous powder, [α]^24^_D_ − 52.5 (c 0.58, MeOH); HR-ESI-MS: *m*/*z*: 851.2216 [M + Na]^+^ (calcd for C_36_H_44_O_22_Na 851.2216); ^13^C-NMR (175 MHz, CD_3_OD) *δ*_C_: 20.8, 20.8 (each CH_3_-CO-), 37.4 (C-5A), 42.8 (C-5B), 45.2 (C-9A), 46.3 (C-9B), 61.9 (C-10A), 62.9 (C-6ʹB), 63.8 (C-10B), 64.4 (C-6ʹA), 71.5 (C-4ʹA), 71.7 (C-4ʹB), 74.6 (C-2ʹA), 74.9 (C-2ʹB), 75.5 (C-6B), 75.8 (C-5ʹA), 77.6 (C-5ʹB), 77.8 (C-3ʹA), 78.5 (C-3ʹB), 86.4 (C-6A), 93.2 (C-1A), 99.9 (C-1ʹA), 100.5 (C-1ʹB), 101.4 (C-1B), 106.3 (C-4A), 108.1 (C-4B), 129.2 (C-7A), 131.8 (C-7B), 144.1 (C-8A), 146.0 (C-8B), 150.2 (C-3A), 155.8 (C-3B), 168.6 (C-11B), 172.2 (C-11A), 172.6 (2 × CH_3_-CO-).

### 3.5. Acid Hydrolysis

Each compound (2 mg) was refluxed individually in 1 M HCL (1.0 mL) at 80 °C for 3 h. The solution was neutralized with Amberlite IRA96SB (OH^−^ form), then it was filtered. The filtrate was evaporated and partitioned between EtOAc: H_2_O mixture (1:1). The aqueous layer was analyzed by HPLC with an amino column [Ashipak NH2P-50 4E, CH_3_CN-H_2_O (3:1), 1mL/min] and a chiral detector (JASCO OR-2090plus). The peak that appeared at *t*_R_ 8.15 min (positive optical sign) supported the presence of D-glucose in the structures of iridoid glucosides (**1**–**4**) [12].

## 4. Conclusions

In summary, the chemical composition of the leaves of *L. verticillatus* was further investigated to lead the isolation of a new iridoid glycoside, lasianoside F (**1**) and three new bis-iridoid glycosides, lasianosides G–I (**2**–**4**), together with four known compounds (**5**–**8**). The structures of isolated compounds (**1**–**8**) were characterized by physical and spectroscopic data analyses, including one-dimensional (1D) and two-dimensional (2D) NMR, IR, UV, and high-resolution electrospray ionization mass spectra (HR-ESI-MS). The absolute configuration of the new compounds was determined by acid hydrolysis and the analysis of the CD cotton effect. 

## Figures and Tables

**Figure 1 molecules-25-02798-f001:**
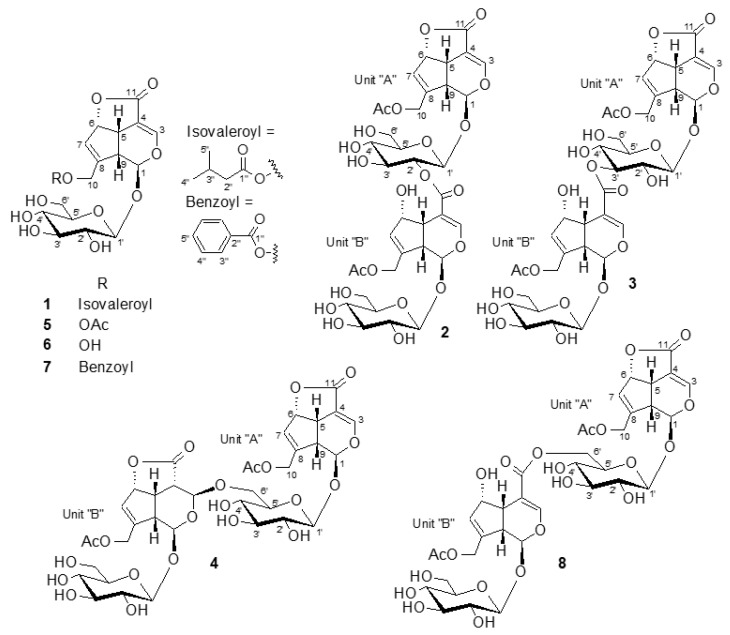
Isolated compounds from *L. verticillatus* (**1**–**8**).

**Figure 2 molecules-25-02798-f002:**
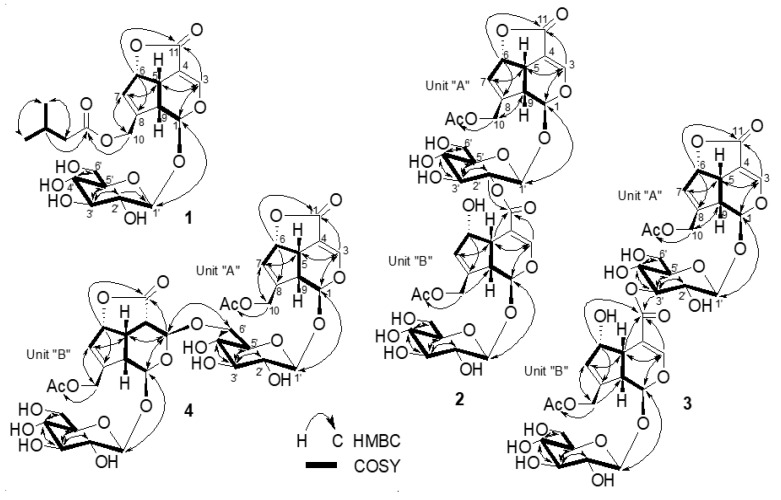
COSY and HMBC correlations of **1**–**4**.

**Figure 3 molecules-25-02798-f003:**
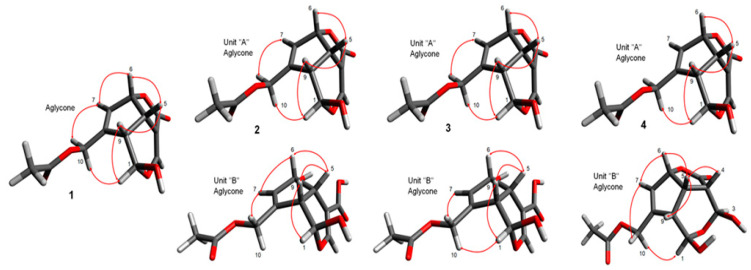
Key NOE correlations of **1**–**4**.

**Table 1 molecules-25-02798-t001:** The ^13^C and ^1^H-NMR spectroscopic data for **1**.

Position	*δ*c	*δ*_H_ Multi (*J* in Hz)
1	93.3	5.98 d (1.0)
3	150.4	7.32 d (2.0)
4	106.2	-
5	37.5	3.70 t-like (6.8)
6	86.4	5.59 br dt (6.5, 1.7)
7	129.1	5.75 br t (1.8)
8	144.4	-
9	45.4	3.31 m
10	61.7	4.69 dd (14.4, 1.0) 4.81 dd (14.4, 1.0)
11	172.6	-
1’	100.0	4.70 d (7.9)
2’	74.7	3.22 dd (9.1, 7.9)
3’	77.9	3.40 br t (8.9)
4’	71.6	3.30 m
5’	78.4	3.37 ddd (9.3, 6.0,2.1)
6’	62.8	3.70 dd (11.8, 6.0) 3.94 dd (11.8, 2.1)
1’’	174.2	-
2’’	44.0	2.28, 2H, d (7.2)
3’’	26.8	2.09 nonet-like (6.8)
4’’/5’’	22.8	0.98, 6H, d (6.6)

m: multiplet or overlapped signals. (175, 500 MHz, CD_3_OD, *δ* in ppm).

**Table 2 molecules-25-02798-t002:** ^1^H-NMR data of compounds **2**–**3** (500 MHz, CD_3_OD, δ in ppm, *J* in Hz).

Position	2		3	
	Unit A *δ*_H_ Multi (*J* in Hz)	Unit B *δ*_H_ Multi (*J* in Hz)	Unit A *δ*_H_ Multi (*J* in Hz)	Unit B *δ*_H_ Multi (*J* in Hz)
1	5.86 d (1.4)	5.05 d (9.1)	5.99 br s	5.07 d (9.2)
3	7.15 d (1.9)	7.70 d (1.1)	7.33 br d (1.7)	7.77 br d (1.3)
4	-	-	-	-
5	3.43 m	2.86 ddd (7.1, 5.7, 1.1)	3.69 m	3.12 br t (6.8)
6	5.51 br dt (6.6, 1.6)	4.80 m	5.59 br t (6.2)	4.89 m
7	5.69 br t (1.6)	6.00 br d (1.7)	5.76 br s	6.05 br d (1.9)
8	-	-	-	-
9	3.27 m	2.73 t-like (8.5)	3.36 m	2.72 t-like (8.1)
10	4.62 dd (14.6, 0.9) 4.74 m	4.82 m 4.95 dd (14.8, 0.6)	4.69 br d (14.4) 4.80 m	4.82 m 4.97 br d (15.8)
11	-	-	-	-
10-COCH_3_	-	-	-	-
10-COCH*_3_*	2.05 s	2.09 s	2.10 s	2.10 s
1’	4.92 d (8.2)	4.70 d (7.8)	4.84 m	4.75 d (7.9)
2’	4.80 m	3.27 m	3.44 dd (9.6, 8.1)	3.27 t-like (8.3)
3’	3.67 t-like (9.2)	3.38 m	5.08 t-like (8.6)	3.30 m
4’	3.38 m	3.29 m	3.59 t-like (9.4)	3.28 m
5’	3.45 m	3.29 m	3.51 ddd (9.8, 5.6, 1.9)	3.40 br d (8.8)
6’	3.69 dd (11.9, 6.7) 3.94 dd (11.9, 1.8)	3.61 dd (12.0, 5.6) 3.83 dd (12.0, 1.4)	3.74 m 3.95 dd (11.9, 1.9)	3.63 dd (11.9, 5.8) 3.87 dd (11.9, 1.7)

m: multiplet or overlapped signals.

**Table 3 molecules-25-02798-t003:** ^13^C-NMR spectroscopic data for **2** and **3** (125, 175*) MHz, CD_3_OD, *δ* in ppm).

Position	2		3 *	
	Unit A	Unit B	Unit A	Unit B
1	94.1	101.8	93.4	101.4
3	150.1	156.3	150.3	156.2
4	106.3	107.7	106.2	108.0
5	37.6	42.9	37.5	42.5
6	86.1	75.2	86.4	75.8
7	129.1	131.6	129.0	131.5
8	143.9	146.5	144.3	146.2
9	45.1	45.9	45.3	46.4
10	61.9	63.8	62.0	63.8
11	172.2	167.6	172.6	168.6
10-COCH_3_	172.7	172.2	172.4	172.6
10-COCH_3_	20.6	20.8	20.7	20.8
1’	98.7	100.9	100.0	100.6
2’	74.5	74.9	73.0	74.9
3’	75.6	77.8	78.7	77.9
4’	71.6	71.7	70.0	71.6
5’	78.5	78.6	78.6	78.6
6’	62.7	63.0	62.4	63.0

*: Measured by 175 MHz.

**Table 4 molecules-25-02798-t004:** The ^13^C and ^1^H-NMR spectroscopic data for **4** (175 MHz, 500 MHz, CD_3_OD, δ in ppm).

Position	Unit A		Unit B	
	*δ* _c_	*δ*_H_ multi (*J* in Hz)	*δ* _c_	*δ*_H_ multi (*J* in Hz)
1	93.5	5.91 d (1.1)	97.0	5.14 d (6.0)
3	150.3	7.33 d (1.9)	97.4	5.27 d (3.6)
4	106.3	-	44.4	3.36 m
5	37.5	3.69 td-like (6.8, 1.8)	37.5	3.47 m
6	86.4	5.59 dt (6.6, 1.4)	87.9	5.41 br d (6.5)
7	128.9	5.75 br s	125.9	6.01 br s
8	144.3	-	152.5	-
9	45.5	3.37 m	46.3	3.05 m
10	61.9	4.68 br dd (14.3, 1.0) 4.79 dd (14.6, 1.2)	62.8	4.68 dd-like (14.3, 1.0) 5.00 br d (15.9)
11	172.6	-	176.9	-
10-COCH_3_	172.2	-	172.6	-
10-COCH_3_	20.8	2.13 s	20.8	2.09 s
1’	100.2	4.75 d (8.1)	99.6	4.73 d (8.1)
2’	74.5	3.24 dd (8.9, 8.1)	75.0	3.24 dd (8.9, 8.1)
3’	77.8	3.42 m	77.9	3.43 m
4’	71.2	3.42 m	71.6	3.30 m
5’	76.6	3.57 br dd (9.4, 3.6)	78.4	3.32 m
6’	68.1	3.95 dd (11.7, 1.4) 4.18 dd (11.7, 5.0)	62.8	3.67 dd (11.7, 3.6) 3.88 br d (11.7)

m: multiplet or overlapped signals.

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
