# Peer review of "Lasianosides F–I: A New Iridoid and Three New Bis-Iridoid Glycosides from the Leaves of Lasianthus verticillatus (Lour.) Merr."

_molecules, 2020, doi:10.3390/molecules25122798_

Round 1

Reviewer 1 Report

COMMENTS:

The manuscript (MS) by Al-Hamoud et al. titled “A new iridoid and three new bis-iridoid glycosides from the aerial parts of Lasianthus verticillatus Hemsl” describes a classical study of the title isolate 1 and three more complex systems 2-4, which were analyzed together with the known compounds 5-8 of the same type. The structures of glycosides 1-4 were established by spectroscopic methods including 1D and 2D NMR experiments in combination with HR-ESI-MS and CD spectra. Their acid hydrolysis followed by the HPLC analysis also supported the presence of the D-glucose molecular units. None of the compounds 1-8 exhibited any significant antioxidant activity or cytotoxicity. The topic of this short work is potentially publishable in Molecules. Hence, I recommend this MS for publication in Molecules after a minor revision, i.e., after considering all of the issues listed below.

Major Point:

1     The problem of a more accurate description of NMR spectra, especially for 1H nuclei, has been widely discussed in a recent important review by McAlpine et al. (Nat. Prod. Rep. 2019, 36, 35-107). It is even recommended to include raw NMR data - and more specifically FID files - when publishing NMR data on new natural products. Regarding the coupling constants nJHH, they are usually given with an uncertainty of ±0.1 Hz to allow later NMR spectral data reproduction, for example by calculation; see Table 1 in the above article.

Some Minor Points:

  1. Row 44: ………(2-4), together ….. compounds (5-8). → ………(2-4), together ….. compounds (5-8).
  2. Figure 1: There is a problem with the number/symbol under compound 4.
  3. Tables 1, 2 and 4: I may opinion, the couplings nJHH can be evaluated from 500 MHz spectra more accurately than with uncertainty of ±0.5 Hz (see, major point 1 above). This should be done.
  4. Rows 94-100: A little too many repetitions of the figures given in Table 1. It should be corrected.
  5. Row 231: perhaps …….500 and 700 MHz spectrometers → ……… 500 MHz spectrometer.

Reviewer 2 Report

The Authors present the isolation and structure elucidation of a new iridoid glycoside and three new bis-iridoid glycosides, together with four known compounds. Overall, the data are presented in a concise and clear manner.

I have some concerns:

  • At least some analytical data should be given for the identity of compound 5-8
  • In my opinion it doesnt make much sense to talk about the biological activity of these compounds if there is none
  • The configuration of the iridoid units was established by NOE. What about the configuration of C1?
  • IUPAC nomenclature for the new compounds is missing
  • There are some strange numbers in Fig. 1 (52-55)
  • Please make sure that there are no page breaks in the figures or tables
  • In the conclusion you state that he absolute configuration of the new compounds was determined by the electronic circular dichroism data which is not in agreement wit acidic hydrolyses and enantioselective HPLC in the discussion?

Reviewer 3 Report

The paper entitled “ A new iridoid and three new bis-iridoid glycosides 2 from the aerial parts of Lasianthus verticillatus 3 Hemsl.” isolates and characterizes 4 new compounds found in Lasianthus verticillatus. From my point of view, this work presents two main problems:

1.In the introduction lack a hypothesis and a justification of why it is necessary to search and identify more compounds. These authors have published an article in Molecules: “Lasianosides A–E: New Iridoid Glucosides from the Leaves of Lasianthus verticillatus (Lour.) Merr. And Their Antioxidant Activity. Molecules 2019, 24, 3995; doi:10.3390/molecules24213995” . Why don't these compounds have been introduced in this article?. The species and the methodology used is the same.

The introduction is deficient. It has little information. Línea 43-44: “leading to isolation of a new iridoid glycoside (1) and three new bis-iridoid glycosides (2-4), together with four known compounds (5-8)”, There is no citation reference to these compounds.

The figures and tables before being cited in the text and even , figure 1 , is not cited in the text. Nor do I think it right that in the introduction to get results (figure 1).

2-There is one aspect that is more worrying: the methodology explained to quantify the antioxidant activity by DPPH is not correct.

-1 μL of different concentrations of the tested compounds were added in 96-well microtiter plate with 100 μL of MeOH, and their absorbances were measured at 515nm as AS0. This is considered as the blank and it is not true, blank is the DPPH.

-% Inhibition = [1 - (AS0 - AS30)/ (AD0 - AD30)] x 100 ; where AD0 and AD30 are the absorbances of DMSO with all reagents solution. To that DMSO is referred to?

-different concentrations of the tested compounds were added in 96-well microtiter plate with 100 μL of MeOH: What concentrations of compounds are used?

This same methodology is explained and used in the article published: Molecules 2019, 24, 3995; doi:10.3390/molecules24213995 and it is not correct.

Round 2

Reviewer 2 Report

Please check Table 1, which appears in line 75 and 76 and separates the paragraph

Reviewer 3 Report

-It is a descriptive work that is reduced to the identification of compounds in a species. The description on chemotaxonomic and biosynthetic information in conclusions are not relevant. The editor must decide if the contribution made with this work is relevant to be published in Molecules.

-“The figures and tables before being cited in the text and even , figure 1 , is not

cited in the text. Nor do I think it right that in the introduction to get results

(figure 1).

Thank you for the valuable comment. We investigated this point. And

added the information of 'Fig' and 'Table' to the appropriate position.”

Figure 5 no

- “There is one aspect that is more worrying: the methodology explained to

quantify the antioxidant activity by DPPH is not correct.

-1 μL of different concentrations of the tested compounds were added in 96-

well microtiter plate with 100 μL of MeOH, and their absorbances were

measured at 515nm as AS0. This is considered as the blank and it is not true,

blank is the DPPH.

-% Inhibition = [1 - (AS0 - AS30)/ (AD0 - AD30)] x 100 ; where AD0 and AD30

are the absorbances of DMSO with all reagents solution. To that DMSO is

referred to?

-different concentrations of the tested compounds were added in 96-well

microtiter plate with 100 μL of MeOH: What concentrations of compounds are

used?

This same methodology is explained and used in the article published: Molecules

2019, 24, 3995; doi:10.3390/molecules24213995 and it is not correct.

This part was removed according to the other reviewer's comment.”

2.2. Antioxidant and Cytotoxic Activities of Isolated Compounds:  no  has not been removed from results

On the other hand, for future work, I recommend the authors to read articles that describe and explain the chemical principle of different methods to quantify antioxidant activity and thus be able to choose the most appropriate method for the molecules that they want to study.
